# Atrophin controls developmental signaling pathways via interactions with Trithorax-like

Kelvin Yeung[1,2], Ann Boija[3], Edvin Karlsson[4,5], Per-Henrik Holmqvist[3], Yonit Tsatskis[1,2], Ilaria Nisoli[6†], Damian Yap[7,8], Alireza Lorzadeh[9,10,11], Michelle Moksa[9,10,11], Martin Hirst[9,10,11], Samuel Aparicio[7,8], Manolis Fanto[6], Per Stenberg[4,5], Mattias Mannervik[3]*, Helen McNeill[1,2]*

[1]Department of Molecular Genetics, University of Toronto, Toronto, Canada; [2]Lunenfeld-Tanenbaum Research Institute, Mount Sinai Hospital, Toronto, Canada; [3]Department of Molecular Biosciences, The Wenner-Gren Institute, Stockholm University, Stockholm, Sweden; [4]Department of Molecular Biology, Umeå University, Umeå, Sweden; [5]Division of CBRN Security and Defence, FOI-Swedish Defence Research Agency, Umeå, Sweden; [6]Department of Basic and Clinical Neuroscience, Maurice Wohl Clinical Neuroscience Institute, King's College London, London, United Kingdom; [7]Department of Molecular Oncology, British Columbia Cancer Research Centre, Vancouver, Canada; [8]Department of Pathology and Laboratory Medicine, University of British Columbia, Vancouver, Canada; [9]Department of Microbiology and Immunology, University of British Columbia, Vancouver, Canada; [10]Michael Smith Laboratories, Vancouver, Canada; [11]Canada's Michael Smith Genome Sciences Centre, BC Cancer Agency, Vancouver, Canada

*For correspondence: mattias. mannervik@su.se (MMa); mcneill@ lunenfeld.ca (HM)

Present address: †Division of Infection and Immunity, University College London, London, United Kingdom

**Abstract** Mutations in human *Atrophin1*, a transcriptional corepressor, cause dentatorubral-pallidoluysian atrophy, a neurodegenerative disease. *Drosophila Atrophin* (*Atro*) mutants display many phenotypes, including neurodegeneration, segmentation, patterning and planar polarity defects. Despite Atro's critical role in development and disease, relatively little is known about Atro's binding partners and downstream targets. We present the first genomic analysis of Atro using ChIP-seq against endogenous Atro. ChIP-seq identified 1300 potential direct targets of Atro including *engrailed*, and components of the Dpp and Notch signaling pathways. We show that Atro regulates Dpp and Notch signaling in larval imaginal discs, at least partially via regulation of *thickveins* and *fringe*. In addition, bioinformatics analyses, sequential ChIP and coimmunoprecipitation experiments reveal that Atro interacts with the *Drosophila* GAGA Factor, Trithorax-like (Trl), and they bind to the same loci simultaneously. Phenotypic analyses of *Trl* and *Atro* clones suggest that Atro is required to modulate the transcription activation by Trl in larval imaginal discs. Taken together, these data indicate that Atro is a major Trl cofactor that functions to moderate developmental gene transcription.

## Introduction

Atrophin family transcription factors are conserved transcriptional corepressors essential for development. Humans have two *Atrophin* genes; *Atrophin 1* (*ATN1*) and *Atrophin 2*. A polyglutamine expansion in *ATN1* is responsible for dentatorubral-pallidoluysian atrophy (DRPLA) a progressive disorder of ataxia, myoclonus, epilepsy, intellectual deterioration and dementia (*Koide et al., 1994*;

**eLife digest** Cells with the same genetic information can look and behave differently to each other. This is because they can control the activity of their genes, changing the effects the genes have in the cell. Regulating genes in this way is important in allowing cells to adapt to their surroundings and to perform different tasks.

Proteins called transcription factors control the activity of genes through other proteins called transcriptional co-activators and co-repressors. Atrophins are a group of co-repressors found in many animals including humans and fruit flies. Atrophins suppress the activity of certain genes, reducing the effects that they have in the cell. Losing Atrophin from cells can lead to severe diseases, but how Atrophin causes these effects is currently not well understood.

Yeung et al. examined which genes Atrophin regulates in cells from the fruit fly *Drosophila melanogaster*. This investigation revealed that, amongst other genes, Atrophin controls several well-studied genes including *engrailed* and *thickveins*. These genes are important in allowing cells to communicate and co-ordinate before birth, ensuring cells work together to build complex tissues and organs. These results suggest Atrophin plays key roles in organising and shaping the body before birth.

Further examination revealed that Atrophin acts in partnership with another molecule called Trithorax-like. Inside the cell many genes are protected by structures called nucleosomes that make them difficult to access, and Trithorax-like helps Atrophin to gain access to these genes. Further work will examine whether Atrophin and Trithorax-like work directly together or if other molecules bring about their interaction. It will also be important to examine how Atrophins suppress the activity of the genes they control. Errors in Atrophin1 in humans result in a nerve-damaging disease known as DRPLA; this work could also help researchers to better understand this disorder.

*Nagafuchi et al., 1994*). *Drosophila* has only one *Atrophin* (*Atro*) gene, and expression of polyglutamine expansion *Atro* also lead to neurodegeneration (*Napoletano et al., 2011*). Loss of *Atro* results in defects in planar polarity, segmentation, and eye, wing and leg developmental defects (*Erkner et al., 2002*; *Zhang et al., 2002*; *Fanto et al., 2003*; *Charroux et al., 2006*; *Napoletano et al., 2011*; *Saburi et al., 2012*; *Sharma and McNeill, 2013*). Atro is required for proper expression of *en* in the embryo and represses the gap genes *Krüppel* and *knirps* (*Erkner et al., 2002*; *Zhang et al., 2002*; *Haecker et al., 2007*). Atro also affects epidermal growth factor receptor (EGFR), Decapentaplegic (Dpp) and Hedgehog signaling (*Erkner et al., 2002*; *Charroux et al., 2006*; *Zhang et al., 2013*). Despite regulating many pathways and processes, only three direct Atro targets have been defined (*knirps*, *fat* and *dpp*) (*Wang et al., 2006*; *Napoletano et al., 2011*; *Zhang et al., 2013*). These targets cannot adequately explain the diverse phenotypes exhibited by loss of *Atro*.

Atro physically interacts with histone deacetylase 1 (HDAC1) and the histone methyltransferase, G9a, through its SANT and ELM2 domains, respectively (*Figure 1A*) (*Wang et al., 2006*, *2008*). These interactions are thought to contribute to Atro's repressor activity. Atro also interacts with other cofactors such as the co-repressor, Scribbler (also called Brakeless) (*Wang et al., 2006*; *Haecker et al., 2007*). Atro does not have a DNA-binding sequence and has been shown to interact with DNA via interactions with nuclear receptors (*Wang et al., 2006*). We show here that a major means by which Atro regulates transcription is with the *Drosophila* GAGA factor, Trithorax-like (Trl, also called GAF). Trl is an essential transcription factor that directly binds to GA repeats (*Biggin and Tjian, 1988*; *Soeller et al., 1988*). Trl regulates many developmentally important genes including the segment polarity gene, *engrailed* (*en*) (*Farkas et al., 1994*; *Bejarano and Busturia, 2004*). Trl can interact with other transcription factors (e.g. Yorkie, Tramtrack) to regulate transcriptional targets in patterning and growth control (*Pagans et al., 2002*; *Oh et al., 2013*).

Trl was initially discovered to bind to the promoter of *Ultrabithorax* (*Ubx*) to activate transcription of *Ubx* in vitro (*Biggin and Tjian, 1988*). Trl is also required in transcriptional repression. This is supported by three pieces of evidence: First, Trl binds to Polycomb response elements, DNA sequences where Polycomb group proteins bind to repress transcription of target genes (*Horard et al., 2000*;

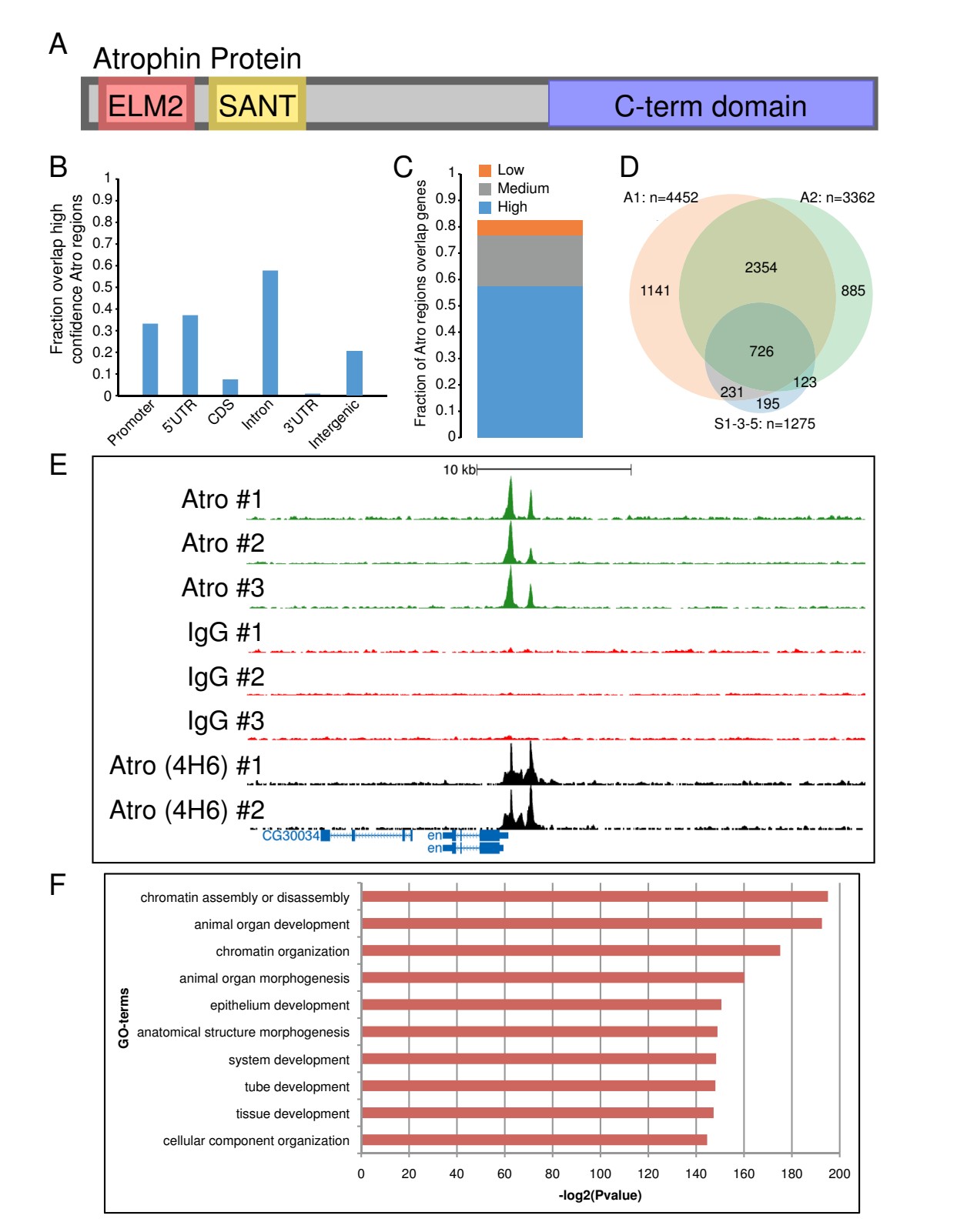

**Figure 1.** Atrophin ChIP-seq results. (**A**) A schematic of Atro protein is shown. Atro's N-terminal side has ELM2 and SANT domains to interact with HDAC1 and G9a, respectively. Atro's C-terminal domain interacts with Tailless and Fat. (**B**) Fraction of the common Atro regions overlapping with various genomic features (genome release 5.57) (**Attrill et al., 2016**). (**C**) The fraction of common Atro regions overlapping genes (including 500 bp upstream of transcription start site) with high, medium or low expression in S2 cells (data from [**Cherbas et al., 2011**]). (**D**) Venn diagram showing

*Figure 1 continued on next page*

*Figure 1 continued*

overlap of the common regions from three biological replicates (S1-3-5) with two additional, independent biological replicates (A1 and A2) using a different Atro antibody (4H6). Only the peaks at the major chromosome arms (heterochromatin are excluded) are compared, and thus, only 1275 peaks are used in S1-3-5 for this analysis instead of 1377 peaks. (E) Example overlap of Atro ChIP-seq peaks at the *engrailed* (*en*) locus. Atro #1–3 are the triplicates Atro ChIP-seq data; IgG #1–3 are the corresponding IgG ChIP-seq controls for the Atro ChIP-seq data. Atro (4H6) #1–2 are the independent Atro ChIP replicates (shown in D). (F) Top 10 GO-term enrichment hits of the Atro ChIP-seq data. GO-term enrichments were done with PANTHER overrepresentation test with default parameters and Bonferroni correction (*Mi et al., 2016*).

The following source data and figure supplement are available for figure 1:

**Source data 1.** PANTHER GO-Term enrichment results (*Mi et al., 2016*) for *Figure 1F*.
**Source data 2.** Overlap of ChIP data sets with the three classes defined in PCA analysis of Atro peaks (modENCODE (downloaded from http://inter-mine.modencode.org/) and CBP [*Philip et al., 2015*]) for *Figure 1—figure supplement 1B*.
**Figure supplement 1.** Principal component analysis of Atro peaks.

*Busturia et al., 2001*). Second, Trl can physically associate with Polycomb Repressive Complex I (*Poux et al., 2001*). Third, *Trl* mutations enhance Polycomb group mutations, indicating that Trl is required for Polycomb repression (*Mahmoudi et al., 2003*). In addition to Trl's ability to regulate transcription, Trl binding leads to open chromatin (*Tsukiyama et al., 1994*) and Trl maintains open chromatin (*Fuda et al., 2015*), at least in part by interacting with chromatin remodeling complexes such as NURF and FACT (*Xiao et al., 2001*; *Shimojima et al., 2003*). Trl is also required for transcriptional pausing (*Lee et al., 2008*; *Fuda et al., 2015*). A working model is that Trl first binds to DNA to maintain open chromatin and later associates with other proteins to activate or repress transcription (reviewed in [*Granok et al., 1995*; *Lehmann, 2004*]).

Here, we identify the genome-wide targets of Atro by chromatin-immunoprecipitation followed by sequencing (ChIP-seq). Our ChIP data show that Atro binds to regulatory regions of *en*, and in the putative regulatory regions of multiple Dpp and Notch signaling components. Our genetic and phenotypic analyses of Atro show that Atro regulates Dpp and Notch signaling, via transcriptional regulation of *thickveins* and likely *fringe*, respectively. We find that Atro negatively modulates En expression, while Trl promotes En expression. Bioinformatic analyses reveal that Atro and Trl ChIP-seq data strongly overlap and sequential ChIP and coimmunoprecipitation experiments confirm Atro and Trl bind to the same loci simultaneously and associate with one another. These data indicate that Atro modulates developmental gene expression via Trl binding. Our results suggest that Trl uses Atro as a cofactor to modulate transcription activation.

## Results

### Identification of genome-wide targets of Atro reveals enrichment in developmental patterning pathways

ChIP-seq against Atro was carried out in *Drosophila* Schneider 2 (S2) cells in three biological replicates (three sets of cells were grown and ChIP'ed on different days). ChIP peaks were called with MACS2 (*Zhang et al., 2008*) for each biological replicate using IgG ChIP-seq as the background model. The resulting three lists (containing 1757, 3064 and 3375 peaks) were intersected and peaks found in all three lists and with summits within 100 bp of each other were selected. The resulting list of 1377 peaks were associated with 1300 unique genes by proximity and treated as potential Atro targets (*Supplementary file 1*). We further mapped the 1377 Atro peaks relative to gene features and gene expression, which showed that the majority of peaks are located in actively transcribed genes, often close to the promoters or in introns (*Figure 1B,C*). In addition, we quantified the enrichment of all chromatin factors mapped by modENCODE (44 proteins and 23 histone modifications) at the Atro peaks and performed principal component analyses of our Atro peaks from major chromosome arms (with heterochromatin excluded, thus 1275 peaks were used). After hierarchical clustering of the significant principal components, we defined three classes (*Figure 1—figure supplement 1A*).

We noticed that Class 2 (blue, *Figure 1—figure supplement 1A and B*) peaks are strongly enriched with Polycomb factors, indicating a potential connection between Atro and Polycomb factors.

To validate these peaks, we also performed independent ChIP-seq in two biological replicates with an antibody raised against a different part of Atro. Peaks were called with MACS2 using input as background model and intersected with the peaks identified above. We compared the peaks from major chromosome arms from each of the ChIP-seqs. A large fraction of the Atro-binding regions identified with the first antibody overlapped the ones found with this second antibody (*Figure 1D,E*).

## Atro modulates developmental gene expression during larval development

GO term enrichment analysis (PANTHER, [*Mi et al., 2016*]) of our ChIP-seq data revealed that Atro targets are enriched in chromatin organization, development and morphogenesis (*Figure 1F*). Interestingly, a potential direct target gene of Atro is *engrailed* (*en*, *Figure 1E*), a critical and conserved regulator of development. There are two strong Atro peaks in the *en* promoter (*Figure 1E*, within 2.4 kb upstream of the transcription start site [*Kassis et al., 1992*]). During embryogenesis, Atro is proposed to repress *en* expression with Even-skipped (*Zhang et al., 2002*). The regulatory region of *en* is complex (*Cheng et al., 2014*), and it is not known if Atro regulates *en* expression in larval stages. Therefore, we generated null mutant clones of *Atro* (*Atro*$^{35}$) in larval imaginal discs and examined the expression of En by immunofluorescence.

We found *Atro*$^{35}$ clones have increased En levels in antennal, wing and leg imaginal discs (*Figure 2A,B*, and data not shown). Interestingly, *Atro*$^{35}$ clones only show increased En levels in clones located in the posterior compartment, where En is endogenously expressed (*Figure 2A–C*). Notably, *Atro*$^{35}$ clones in the anterior compartment cannot induce ectopic expression of En (*Figure 2C*). These data suggest that Atro negatively modulates En expression levels to maintain moderate expression in imaginal discs.

## Atro directly regulates Dpp signaling via thickveins

*Atro* mutant clones exhibit phenotypes similar to Dpp signaling defects such as leg patterning defects and expanded wing veins (*Erkner et al., 2002*; *Zhang et al., 2002*), suggesting Atro regulates Dpp signaling. Interestingly, our ChIP-seq data show that several Atro peaks occur in putative regulatory regions of several Dpp signaling pathway components (e.g. *thickveins*, *Daughters against dpp* and *schnurri*, *Figure 3A* and *Supplementary file 1*), suggesting Atro regulates Dpp signaling by directly regulating the expression of Dpp signaling pathway components. Thickveins (Tkv) is a critical Dpp receptor, and *Atro*$^{35}$ clones show upregulation of the *tkv-LacZ* reporter in the wing (*Wehn and Campbell, 2006*). To confirm that Atro represses *tkv* expression, *tkv* transcript levels were assessed in *Atro* knocked down wing discs using in situ hybridization. Indeed, *tkv* expression is increased if *Atro* is knocked down (*Figure 3B*), confirming Atro represses *tkv* expression in the wing.

Atro binds within *tkv*'s gene locus and represses *tkv* transcription; however, it is not clear if this repression is important for the regulation of Dpp signaling. Mothers against dpp (Mad) is a downstream component of Dpp signaling, that is phosphorylated when the cell receives the Dpp signal (*Newfeld et al., 1997*). Phosphorylated Mad (pMad) is found in a broad stripe around the Dpp source and is used as a read-out for Dpp signaling. We stained *Atro*$^{35}$ clones with anti-pMad antibodies to assess Dpp signaling. *Atro*$^{35}$ clones that are found in the endogenous pMad regions have increased pMad levels along the interior border of the clones that is closest to the centre of the discs, where the endogenous Dpp source is located (*Figure 3C*). This pattern of increased pMad expression is expected if Tkv levels are increased in the clone, since increased Tkv levels in *Atro*$^{35}$ clones would lead to an increase in Mad phosphorylation, as these cells will receive more Dpp signal compared to the adjacent non-clone cells (see *Figure 3—figure supplement 1B*). *Atro*$^{35}$ clones also have increased Optomotor blind (Omb, a downstream target of Dpp signaling) expression (data not shown), indicating increased Dpp signaling in the clones (*Grimm and Pflugfelder, 1996*). However, *Atro*$^{35}$ clones far from the Dpp source do not induce ectopic Mad phosphorylation, suggesting Dpp levels are not increased in *Atro*$^{35}$ clones (*Figure 3D*, arrow heads). Consistent with this, *Atro*$^{35}$ clones in the anterior or posterior compartments do not cause ectopic Dpp expression (*Figure 3E*). Our data indicate that Atro directly regulates *tkv* expression and thereby Dpp signaling.

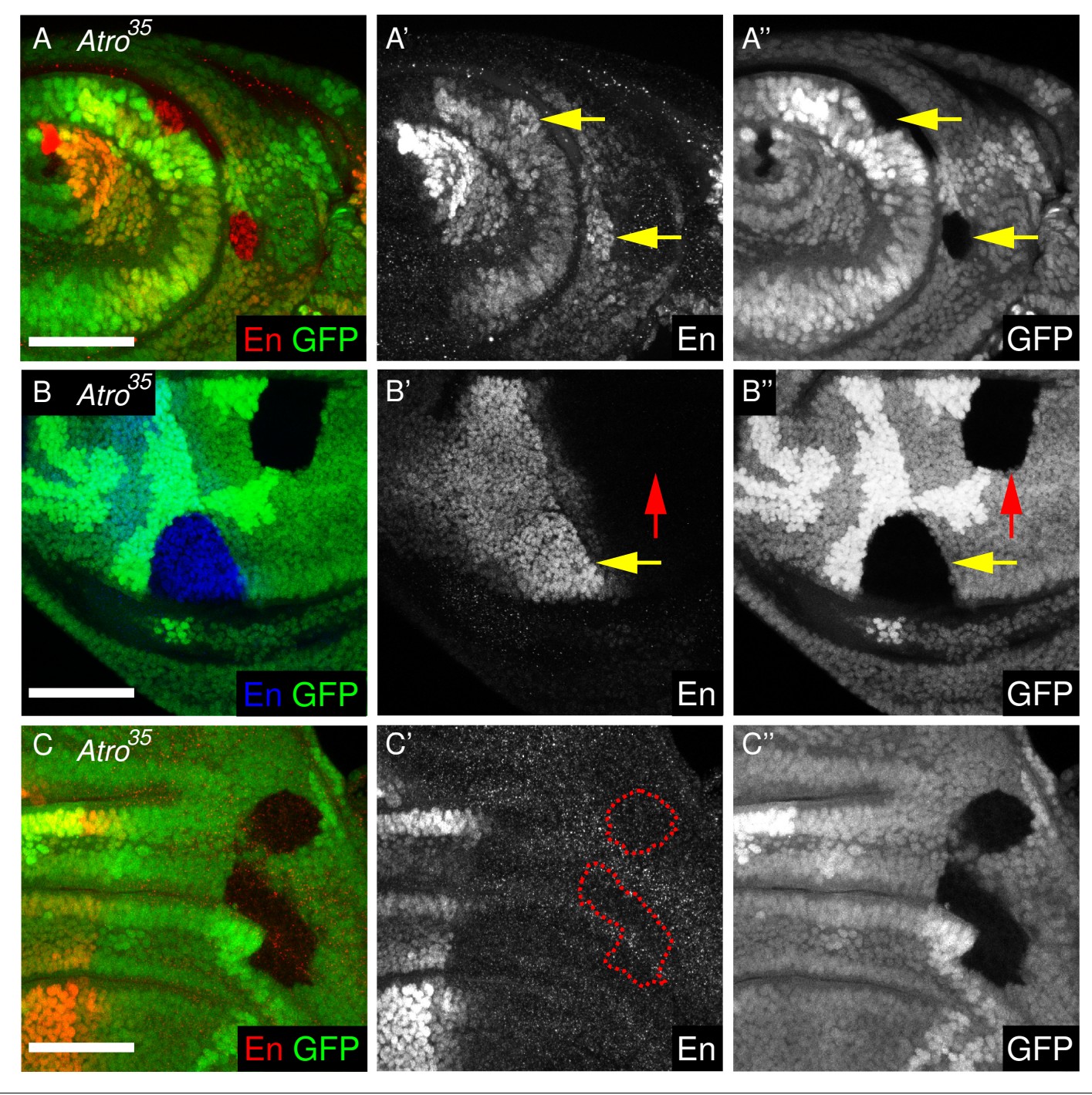

**Figure 2.** Atro is required for modulating En levels in larval imaginal discs. (**A**) Antennal disc clones of *Atro*[35] have increased En levels (arrows). (**B**) *Atro*[35] wing disc clones in the posterior compartment have increased En levels (yellow arrow). But *Atro*[35] clones cannot induce ectopic En expression in the anterior compartment (red arrow). (**C**) *Atro*[35] clones cannot induce ectopic En expression (dotted lines mark clonal borders, wing disc is shown here). (**A**) has posterior to the right; (**B–C**) have posterior to the left, dorsal up. All scale bars are 50 μm.

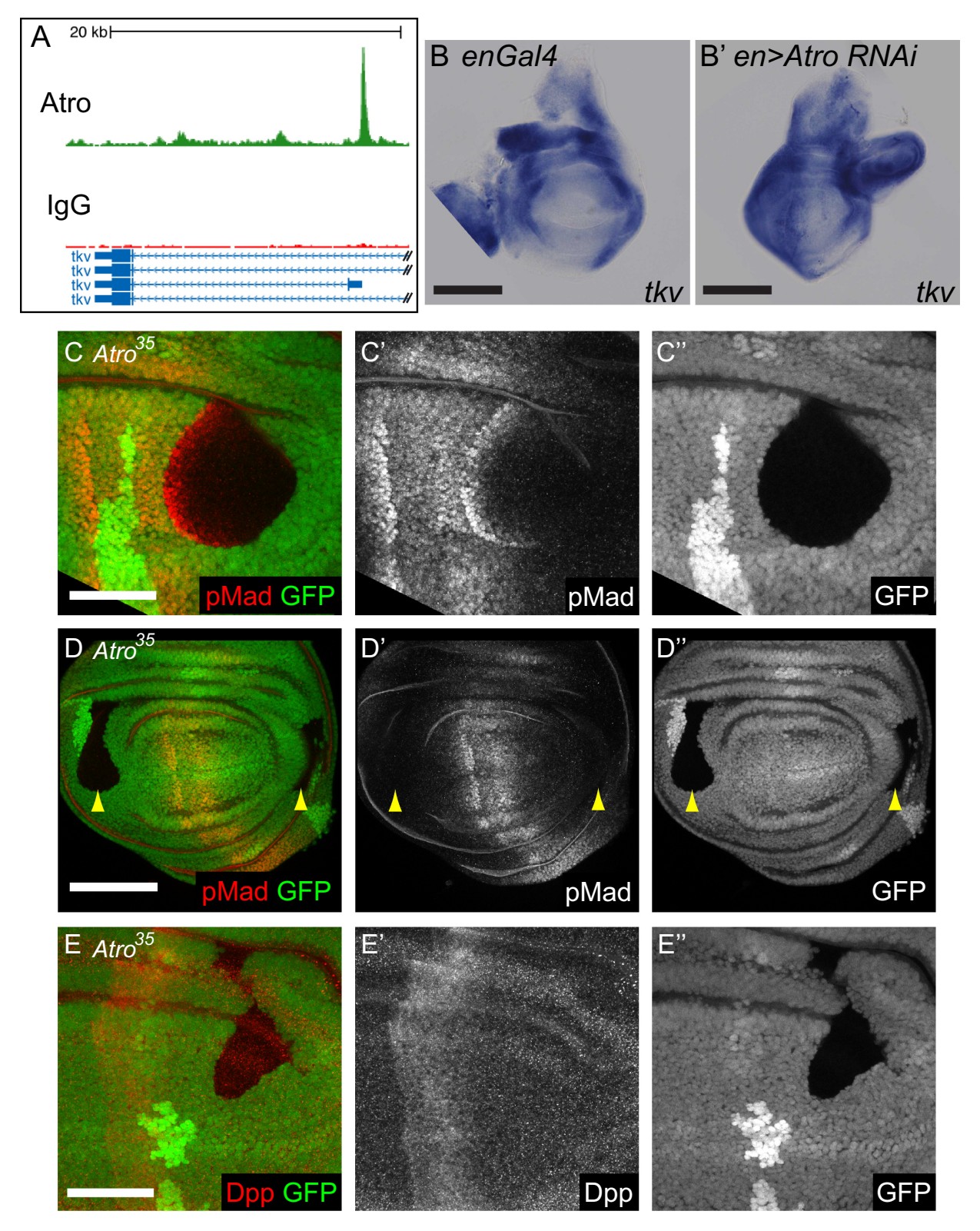

**Figure 3.** Atro regulates Dpp signaling. **(A)** An Atro ChIP peak is found inside the *tkv* locus. This peak is directly upstream of the transcription start site of *tkv* isoform D. **(B)** In situ hybridization showing *enGal4* control wing disc with normal *tkv* expression. **(B')** shows in situ hybridization of *Atro RNAi* driven in the posterior half of the wing disc by *enGal4*. *tkv* expression is increased in the posterior half. **(C)** *Atro*$^{35}$ wing disc clones that cross the endogenous pMad regions have increased pMad levels along the interior border of the clone that is closest to the middle of the disc (where the Dpp

*Figure 3 continued on next page*

*Figure 3 continued*

source is located). This indicates that *Atro$^{35}$* clones have increased Dpp signaling. (D) *Atro$^{35}$* wing disc clones cannot cause ectopic Mad phosphorylation (yellow arrow heads). (E) *Atro$^{35}$* wing disc clones cannot induce ectopic Dpp. All clones are marked by the absence of GFP; all figures have posterior to the left, dorsal up. Scale bars in A and C are 100 μm; in B and D are 50 μm.

The following figure supplement is available for figure 3:

**Figure supplement 1.** Model of how loss of Atro and Trl affect Dpp signaling in wing discs.

## Atro is required for Notch signaling and directly regulates Notch pathway gene expression

Atro ChIP data also revealed that Atro binds the putative regulatory regions of several Notch signaling components (*mastermind* (*mam*), *Delta* (*Dl*), *neuralized* (*neur*) and *fringe* (*fng*)) (***Figure 4A***, ***Supplementary file 1***), suggesting that Atro may regulate Notch signaling. Previous studies had not indicated that Atro loss of function affected Notch pathway activity. Therefore, to explore the biological relevance of the link with Notch, we first tested for genetic interactions between *Atro* and *Notch* (*N*). *N* is required for the development of the wing margin, and heterozygous *N$^{264-39}$* mutant (a null allele of *N*) flies have wing notches in the adult wing (***Figure 4B'***). We find that transheterozygous *N$^{264-39}$*/+; *Atro$^{35}$* /+ mutant flies exhibit strikingly more severe wing notching than heterozygous *N$^{264-39}$* mutant alone (***Figure 4B''***). No notching is observed in *Atro$^{35}$* /+ mutant flies. Similar results were obtained with another independent *Atro* allele, *Atro$^{i5A3}$* (a P-element insertion allele, data not shown). Thus, *Atro* genetically interacts with *N*, suggesting Atro may play a role in N signaling.

Since *Atro* and *N* genetically interact, we decided to test if loss of *Atro* affects wing margin development. In the wing, a line of N signaling induces the expression of wing margin markers Wingless (Wg) and Cut (Ct) (***de Celis et al., 1996***). N signaling in the wing can also be visualized with a N signaling reporter (NRE-GFP) (***Housden et al., 2012***), which expresses in a sharp line in the developing wing margin. *Atro$^{35}$* clones that cross the wing margin disrupt the expression pattern of the NRE-GFP reporter (***Figure 4C***), resulting in diffuse expression of the NRE-GFP reporter. In addition, *Atro$^{35}$* clones that cross the wing margin cause a loss of wing margin markers (***Figure 4D*** and ***Figure 4—figure supplement 1A***). In some cases, *Atro$^{35}$* clones induce ectopic Ct expression just outside of the posterior clonal border (***Figure 4D***, arrow). In contrast to the wing, *Atro$^{35}$* clones do not affect Ct expression in the antennal discs, where a requirement of N signaling for Ct expression has not been shown (***Figure 4—figure supplement 1B***). Atro binds to the *fng* promoter (***Figure 4A***, [***Yang et al., 2013***]), and in situ hybridization shows *Atro* knockdown causes increased *fng* transcription (***Figure 4—figure supplement 1C***). These observations suggest that Atro may be regulating *fng* and/or other factors to affect N signaling.

We also checked if Atro regulates N signaling in the eye. In larval eye discs, R8 photoreceptors differentiate from a three-cell equivalence group, which differentiates into R2, R5 and R8 photoreceptors (reviewed in [***Frankfort and Mardon, 2002***; ***Tsachaki and Sprecher, 2012***]). N signaling is required for lateral inhibition in this equivalence group, such that one cell receives the least amount of N signaling and differentiates into R8. Thus, N signaling loss of function mutants have extra R8's. To see if Atro affects R8 development, we stained *Atro$^{35}$* clones with Senseless (Sens), an early R8 marker. *Atro$^{35}$* clones have extra Sens positive cells (***Figure 4E***), clustered together in groups of two to three cells, resembling the number and shape of the R2/5/8 equivalence group. This suggests that lateral inhibition is defective in *Atro$^{35}$* clones. However, we noticed that there is one cell that has more Sens staining than the rest within each Sens-positive cell cluster (***Figure 4E***, arrowheads). Although *Atro$^{35}$* clones can induce extra Sens-positive cells, these clones do not have an excess of cells marked with a late R8 marker, Bride of sevenless (Boss) (***Figure 4—figure supplement 2A***). In addition, N signaling is required for the differentiation of R7 photoreceptors and cone cells, and loss of N signaling results in a lack of R7 and cone cells in the eye (***Cooper and Bray, 2000***; ***Flores et al., 2000***; ***Tomlinson et al., 2011***). We found that *Atro$^{35}$* clones also lose R7 and cone cell markers (***Figure 4—figure supplement 3***). Extra macrochaetae (*emc*), a downstream target of N signaling (***Bhattacharya and Baker, 2009***), is another potential direct target of Atro (***Supplementary file 1***). Indeed, *Atro$^{35}$* clones have reduced Emc protein levels anterior to the morphogenetic furrow

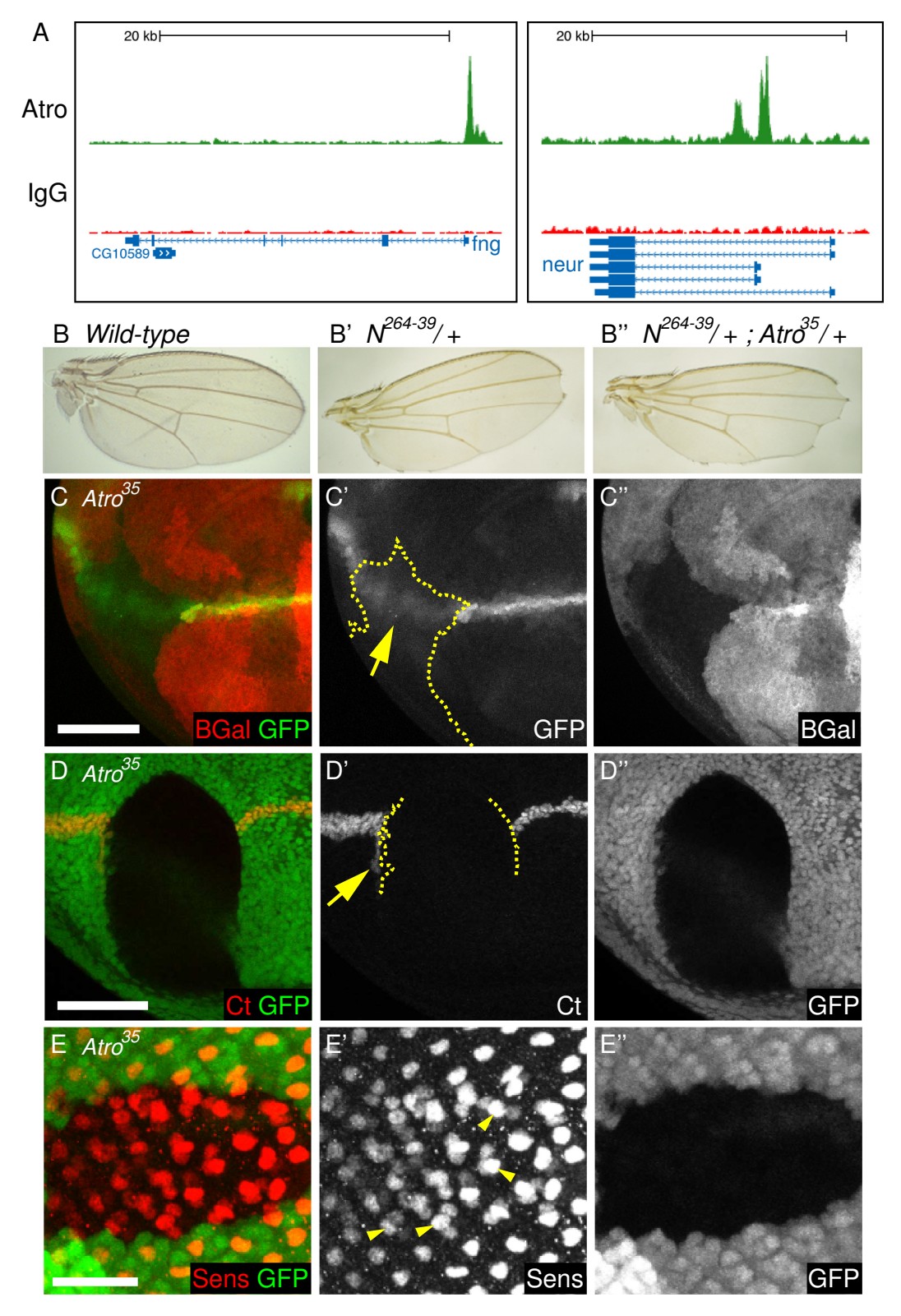

**Figure 4.** Loss of *Atro* leads to loss of Notch signaling phenotypes.  (A) Atro ChIP peaks are found in the *fng* and *neur* loci, indicating that Atro may regulate these genes during development. (B) Wild-type wings have no notches at the wing margin. $N^{264-39}$ heterozygous wings have wing notches (B'). Wing notch phenotype is enhanced in transheterozygous $N^{264-39}$, $Atro^{35}$ wings (B''). (C) $Atro^{35}$ clones cause the Notch signaling reporter (GFP) to express in a diffused pattern (arrow). Dotted line marks clonal borders. (D) Large $Atro^{35}$ clones cause an autonomous loss of wing margin marker Cut as

*Figure 4 continued on next page*

*Figure 4 continued*

well as ectopic Cut expression on the posterior side of the clone (arrow). Clonal borders are marked with dotted lines. (**E**) *Atro35* clones have extra cells expressing the early R8 marker, Sens. Extra Sens—positive cells are found in clusters of two to three cells but one cell within each cluster has more Sens staining than the rest (arrowheads). Clones are marked by absence of ß-Gal (red) in C and clones are marked by the absence of GFP in D and E. B have posterior to the bottom and C to E have posterior to the left. C and D have dorsal side up. Scale bars in C and D are 50 µm; in E is 25 µm.

The following figure supplements are available for figure 4:

**Figure supplement 1.** Loss of Atro causes loss of wing margin markers via Notch signaling.

**Figure supplement 2.** *Atro35* clones have normal Boss expression but altered Emc levels.

**Figure supplement 3.** Loss of Atro have eye phenotypes similar to loss of Notch signaling.

---

(*Figure 4—figure supplement 2B*). Since Atro peaks are present in/very close to multiple N signaling components and downstream targets (*fng*, *numb*, *Dl*, *neur*, *emc*, *Figure 4A*, *Supplementary file 1*), Atro may directly regulate multiple targets to affect N signaling in the eye.

## Atro is a cofactor of Trl

Although Atro does not bind to DNA directly, it binds with other cofactors to associate with DNA (e.g. Tailless [*Wang et al., 2006*]). We reasoned that analysis of the DNA sequences obtained from the Atro ChIP-seq could identify novel potential cofactors and binding motifs of Atro to gain insight into how Atro regulates developmental signaling. Therefore, we performed de novo motif analysis of our ChIP-seq data using MEME-ChIP (*Bailey et al., 2009*) (*Figure 5A*, *Figure 5—figure supplement 1*). The top motif discovered is a GA repeat, the binding motif of Trithorax-like (Trl, *Figure 5A*). The Trl binding motif is enriched within Atro ChIP peak summits, indicating strongest Atro binding. Other enriched motifs include Twin of eyeless (Toy) and Mad (*Figure 5A*). Next, we looked for overlap between the Atro ChIP-seq data with other ChIP datasets from S2 cells. We reasoned that we should see an increase of overlap between Atro and other ChIP data if we looked more specifically at genomic locations with higher Atro ChIP signal using Atro peaks from major chromosome arms (excluding heterochromatin). Therefore, we plotted the amount of Atro overlap with other transcription factors over that expected by chance against the number of genomic locations grouped according to the amount of Atro binding (using genomic data from modENCODE (downloaded from http://intermine.modencode.org/), CBP [*Philip et al., 2015*], and Yki [*Oh et al., 2013*]). In this analysis, several factors, including Trl, CBP (Nejire), the replication proteins Orc2 and MCM, Yorkie (Yki) and Polycomb proteins, showed increased overlap with increasing Atro binding (*Figure 5B*). Interestingly, the strongest overlap was found with Trl, confirming the MEME-ChIP analysis (*Figure 5A*). We further compared our list of Atro peaks with the published Trl ChIP-seq data (*Fuda et al., 2015*) and found striking overlap between the two data sets (1123/1377 Atro peaks overlap with Trl ChIP, *Figure 5C*). Visual inspection of Atro and Trl ChIP-seq peaks also shows binding at the same genomic regions (*Figure 6C,E*).

## Atro and Trl bind simultaneously and form a complex

The above analyses suggest that Atro and Trl bind to the same sites and may form a protein complex. To determine if Atro and Trl form a protein complex, we performed coimmunoprecipitation (coIP) of Atro and Trl in S2 cells. Although we did not detect an interaction using standard coIP conditions, pretreating lysates with micrococcal nuclease allowed us to coimmunoprecipitate endogenous Trl with endogenous Atro (*Figure 5D*).

To directly test if Atro and Trl bind to the same loci simultaneously, we performed sequential ChIP (ChIP-re-ChIP) on Atro followed by Trl in S2 cells. For this analysis, we selected Atro peaks at the *scribbler* (*sbb*) locus because of the presence of an Atro peak that overlaps with a Trl peak as well as a nearby Atro peak that does not overlap (*Figure 6A*). qPCR show that Atro-Trl ChIP-re-ChIP enriches for the locus with both Atro and Trl peaks, while it does not enrich for a locus with only Atro peak (enrichment is nearly equal to Atro-IgG ChIP-re-ChIP-negative control) (*Figure 6B*).

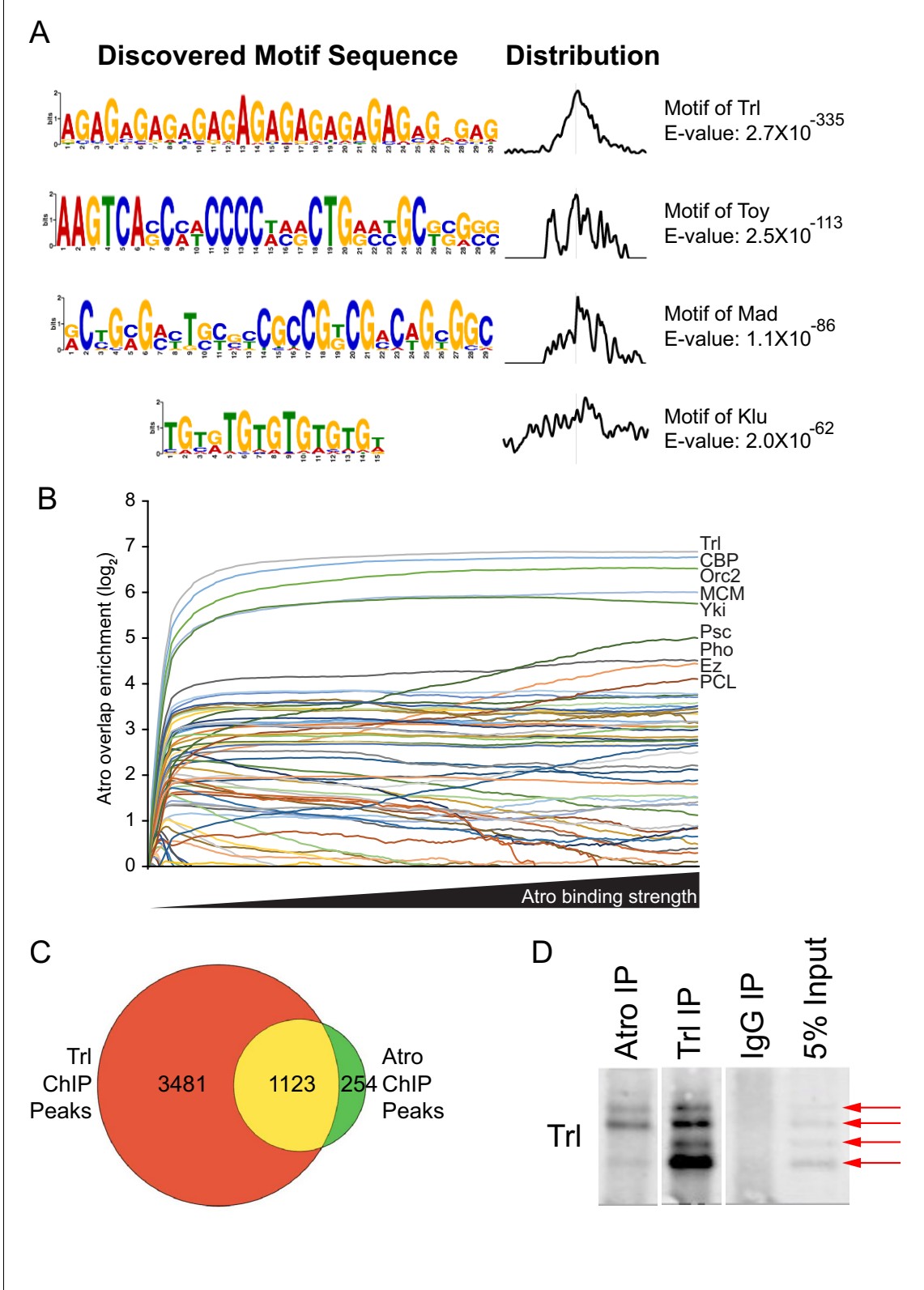

**Figure 5.** Atro and Trl bind to the same genomic locations. (**A**) The top four hits from MEME-ChIP analysis using all Atro peaks. The top hit is a (GA) repeat, which is the binding motif of Trl. (**B**) Atro ChIP-seq data overlap with other factors in S2 cells. The y-axis represents the overlap over that expected by chance, where the fraction of the genome covered by each factor is the overlap expected by chance. Only positive values are shown. The Atro values are selected with increasing cut-off, so that fewer but stronger Atro-binding sites are shown along the x-axis. (**C**) shows the Venn diagram of

*Figure 5 continued on next page*

*Figure 5 continued*

the overlap of all Atro and Trl ChIP data. All 1377 Atro ChIP peaks were used and Trl ChIP peaks are from *Fuda et al. (2015)*. Genomic coordinates from both data sets were used to construct this diagram.

The following source data and figure supplements are available for figure 5:

**Source data 1.** List of Atro peaks' average –log(pvalue) and fold enrichment generated from MACS2 for *Figure 5—figure supplement 2*.
**Figure supplement 1.** Additional MEME ChIP hits of Atro ChIP-seq.
**Figure supplement 2.** Comparison between Atro peaks that overlap Trl and those that do not.

Additional experiments showed that Atro and Trl co-occupy all the tested loci with both Atro and Trl peaks (*Figure 6—figure supplement 1*).

To see if Atro requires Trl to associate with DNA, we knocked down Trl using RNAi in S2 cells. Unfortunately, we could not directly test if Atro association with DNA requires Trl in cells, as knock down of Trl also leads to reduced abundance of Atro proteins (*Figure 6—figure supplement 2A*). These findings indicate that Atro and Trl bind to the same loci and to one another, forming a complex on chromatin to regulate transcription.

## Trl is required for En and Tkv expression

Both ChIP-seq and ChIP-re-ChIP data show that Atro and Trl bind to the same loci simultaneously. Interestingly, there are strong overlaps of Atro and Trl ChIP-seq peaks at the *en* locus (*Figure 6C*). Since we found that Atro negatively modulates the expression of En during larval development, we wondered if Trl also regulates En expression. Therefore, we generated and analyzed *Trl* null mutant clones (*Trl$^{R85}$*) in imaginal discs. In contrast to *Atro$^{35}$* clones, *Trl$^{R85}$* clones have reduced En expression (*Figure 6D*), indicating that Trl is required for the positive regulation of En expression in larval discs.

Atro and Trl ChIP-seq data also indicate that Atro and Trl bind to the same region of the *tkv* locus (*Figure 6E*), suggesting that Trl regulates *tkv* expression with Atro. Therefore, *Trl$^{R85}$* clones were generated to investigate the role of Trl in *tkv* regulation. pMad levels were used as a read out for Tkv levels. In contrast to *Atro$^{35}$* clones, *Trl$^{R85}$* clones have reduced pMad levels (*Figure 6F*, *Figure 6—figure supplement 3*). In addition, pMad staining can be found on the side of the *Trl$^{R85}$* clones further away from the middle of the wing (*Figure 6—figure supplement 3*, arrow), indicating Dpp expression was not disrupted. These changes can be explained by a decrease in Tkv levels in *Trl$^{R85}$* clones. In addition, we did not find any Trl or Atro peaks within 20 kb of the *mad* locus (data not shown). Thus, the simplest model to explain our data is that the reduced pMad in *Trl$^{R85}$* clones is caused by a downregulation of *tkv* expression. Therefore, similar to Atro and Trl's regulation of *en*, Trl is required for *tkv* expression, while Atro is required for repression of *tkv*.

## Discussion

Mutations in Atrophins lead to neurodegeneration, and loss of Atro leads to major defects in development. However, only a direct few targets of Atro are known and it is unclear how Atro regulates many genes. Here, using genome-wide ChIP-seq and in vivo analyses we show that Atro directly binds a number of developmentally critical genes; specifically regulating En expression and Dpp and Notch signaling. Trl has been proposed to activate transcription by opening and maintaining open chromatin to allow additional transcription factors to bind (reviewed in [*Granok et al., 1995*; *Lehmann, 2004*). We find that Atro and Trl participate in a complex and bind to the same loci simultaneously. Atro does not bind to DNA directly, while Trl has a DNA binding domain (*Biggin and Tjian, 1988*). Thus, our results suggest a model in which Atro binds to DNA via Trl, either directly or indirectly via someunknown cofactors, and Atro modulates transcriptional activation by Trl.

Our in vivo analyses of *en* and *tkv* show that Trl is required to promote their expression while Atro negatively modulates expression levels. To explain these phenotypes, we propose a model in which Trl binding leads to open chromatin to allow transcription, and Atro binding, either by directly interacting with Trl or through some unknown factors, negatively modulates Trl's activity

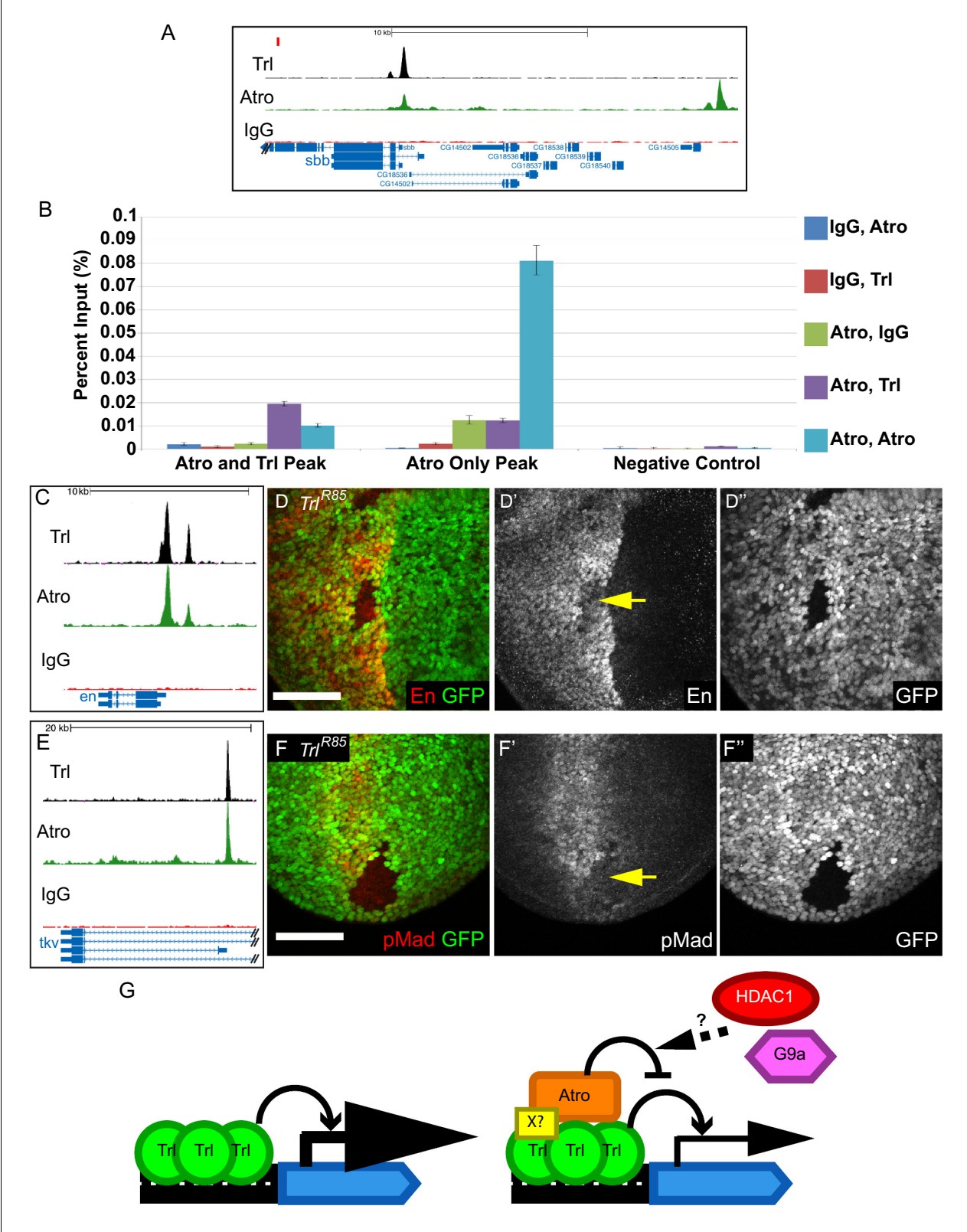

**Figure 6.** Trl and Atro bind to the same loci simultaneously and Trl is required for the expression of *en* and *tkv*. (**A**) There is an Atro peak that overlaps with Trl peak within the *sbb* locus (left peak, Atro and Trl Peak in B). There is an Atro peak that does not overlap with Trl about 20 kb upstream of the *sbb* locus (right peak, Atro Only Peak in B). The red rectangle marks the region used as negative control in B. (**B**) ChIP-re-ChIP qPCR results. ChIP-re-ChIP samples are labeled as the sequence of antibody used (e.g. IgG, Atro means rabbit IgG ChIP followed by Atro re-ChIP). Negative controls for the

*Figure 6 continued on next page*

*Figure 6 continued*

ChIP-re-ChIP are any ChIPs with IgG. Atro, Trl ChIP-re-ChIP enriches the Atro and Trl Peak (purple bar) but it does not enrich the Atro only peak (same enrichment as Atro, IgG). Mean Ct value was used to calculate percent input and standard deviation of the Ct values was carried over in calculations and used as error bars. (C) Trl and Atro ChIP peaks coincide at the same loci upstream of *en*. (D) *Trl^R85* imaginal disc clones have decreased En levels (arrow, wing disc clone shown here). (E) Trl and Atro ChIP peaks coincide at the *tkv* locus. (F) *Trl^R85* wing disc clones have decreased pMad levels (arrow), possibly due to a decrease of *tkv* expression. All clones are marked by the absence of GFP; all figures have posterior to the left, dorsal up. Scale bars are 50 μm. (G) Model of how Atro and Trl function together to regulate expression of target genes. Trl is required to activate the transcription of its target genes. In the absence of Atro (Left), the target gene will express at a higher level than normal. Atro binds to the same site as Trl either directly or via some unknown cofactors (X?, Right). Atro modulates the expression of its target gene by counteracting Trl; potentially Atro is doing so by recruiting Histone deacetylase 1 (HDAC1) and G9a, a histone methyl transferase. Thus, target genes are expressed at the correct levels.

The following source data and figure supplements are available for figure 6:

**Source data 1.** Source data for qPCR %Input calculations for *Figure 6B* and *Figure 6—figure supplement 1*.

**Figure supplement 1.** Additional ChIP-re-ChIP qPCR results of the *Trl* and *no ocelli* (*noc*) loci.

**Figure supplement 2.** Trl knockdown decreases Atro protein levels, coimmunoprecipitation of Trl and Atro, and pausing indices of Trl and Atro bound genes.

**Figure supplement 3.** *Trl^R85* clones cause an autonomous decrease Mad phosphorylation but do not affect pMad levels outside clone (arrow).

(*Figure 6G*). Atro binds to Histone deacetylase 1 (HDAC1) and a histone methyltransferase (G9a) to close chromatin and repress transcription (*Wang et al., 2006*, *2008*), suggesting a mechanisms by which Atro is able to counteract Trl. Indeed, Atro- and Trl-binding sites at the *en*, *tkv* and *fng* loci overlap with HDAC1-binding sites (from modENCODE, RPD3-Q3451) (*Celniker et al., 2009*). Thus, loss of Trl leads to loss of expression of *en* and *tkv* where they are normally expressed. In the absence of Atro, Trl's activity is no longer negatively modulated, and expression of target genes such as *en* and *tkv* are increased. However, in regions where *en* and *tkv* are not endogenously expressed, loss of Atro cannot induce ectopic expression as Atro's function is to negatively modulate the transcription of its target genes.

Interestingly, Trl is enriched at paused promoters and is involved in transcription pausing (*Lee et al., 2008*; *Fuda et al., 2015*). Trl recruits the negative elongation factor NELF to paused genes (*Li et al., 2013*), but the precise mechanism of how Trl affects pausing is unknown. Our data suggest that Atro is present to modulate Trl's transcription activation role. Perhaps, Atro is part of the transcriptional machinery used by Trl to pause transcription. In support of this, we noted that Atro preferentially binds to genes with higher expression (in S2 cells, genes with high expression are likely to be paused, *Figure 1C*) (*Gilchrist et al., 2010*). GO analysis of Atro targets shows enrichment of signaling and patterning genes, also consistent with a role in pausing (*Figure 1F*, data not shown). Notably, the pausing index for genes bound by both Atro and Trl is higher on average than for all expressed genes (*Figure 6—figure supplement 2C*, [*Kwak et al., 2013*]). This increase of pausing index is even higher than genes bound only by Trl (*Figure 6—figure supplement 2B*). These data suggest that Atro may be involved in transcriptional pausing with Trl.

We note that there are some Atro-binding sites that are not co-occupied by Trl. Atro also binds to other transcription factors (such as Tailless and Cubitus Interruptus [*Wang et al., 2006*; *Zhang et al., 2013*]) and Atro may interact with other factors to bind to sites not co-occupied by Trl. Visual inspection reveals strong Atro peaks that do overlap with Trl (*Figure 6A*); however, on a genome-wide basis, Atro peaks that do not overlap with Trl peaks are slightly weaker than the ones that do, as judged from the MACS2 derived significance (-log(p-value)) and fold enrichment values (*Figure 5—figure supplement 2*).

Atro maternal mutants have missing, malformed and/or expanded En stripes (*Zhang et al., 2002*), while in discs Atro modulates En expression. A possible explanation is that En regulation changes at different developmental times. During early embryogenesis, Pair-rule genes are required to first establish *en* expression (*DiNardo et al., 1988*). Atro is also required for the proper expression of Pair-rule genes such as *fushi tarazu* and *even-skipped* (*Erkner et al., 2002*; *Zhang et al.,*

*2002*). This could explain why loss of Atro during early embryogenesis has more severe effects on *en* expression. However, from late embryogenesis on, En expression no longer requires Pair-rule genes but depends on Polycomb group genes and unidentified activators (*Moazed and O'Farrell, 1992*). We observed reduced En expression in *Trl* clones in antennal and wing imaginal discs, while *Atro* disc clones have increased expression. These results suggest that Trl and Atro are required for regulation of En expression from late embryogenesis through larval development.

Why is it important for Atro to moderate *en* expression? Intriguingly, expressing higher than normal levels of En in the posterior compartment leads to lethality and anterior-posterior patterning defects of the posterior wing, suggesting that moderating En levels is required for normal development, and high levels could be toxic (*Guillén et al., 1995*; *Tabata et al., 1995*).

Our Atro ChIP data show that Atro binds to the *tkv* locus and our in situ analysis reveals Atro represses *tkv* expression. We find that *Atro*[35] clones have increased Dpp signaling, consistent with Atro's role in *tkv* regulation. Conversely, *Trl*[R85] clones have lower pMad levels. We reason that this is caused by decreased Tkv within the *Trl*[R85] clones, consistent with a peak in the *tkv* locus. These observations mirror what we have seen with *en*, where Trl is required for activation and Atro is required for repression of transcription.

A model for Atro and Trl regulation of Dpp signaling is shown in *Figure 3—figure supplement 1*. In wild-type wings, the expression patterns of Dpp (yellow shading) and Tkv (red line) cause pMad to be found in a broad stripe (*Figure 3—figure supplement 1A*, blue line) (*Tanimoto et al., 2000*). Tkv levels are increased in *Atro*[35] clones (*Figure 3—figure supplement 1B*, indicated by the shaded rectangle). Thus, pMad levels are increased along the interior border of the clone if the *Atro*[35] clones are close enough to the Dpp source. *Atro*[35] clones further away would not cause changes to pMad levels. Trl has the opposite effect where *Trl*[R85] clones (*Figure 3—figure supplement 1C*, shaded rectangle) cause a decrease of Tkv and thus lower pMad levels. Interestingly, the Mad binding motif is enriched in the Atro ChIP, with a shift in distribution, suggesting Mad may bind adjacent to Atro, and that Atro-Mad interactions on chromatin may also affect Dpp signaling.

Although *Atro*[35] clones can lead to ectopic *dpp-LacZ* expression (*Erkner et al., 2002*; *Zhang et al., 2013*), we did not see increased Dpp in *Atro*[35] wing clones. Additionally, loss of Atro away from the endogenous Dpp stripe does not induce ectopic pMad. Therefore, the increase of pMad staining in *Atro*[35] clones matches the pattern that is expected if Atro regulates Dpp signaling via *tkv* and not via *dpp* regulation. Thus, we suggest that Atro regulates Dpp signaling in the wing primarily by regulating *tkv* transcription.

To our knowledge, this report also provides the first evidence that Atro regulates N signaling. Our ChIP-seq analysis revealed Atro binding in multiple N pathway components, and genetic and molecular analysis reveals disruption of N signaling in Atro clones in eye and wing discs. *Atro* knockdown results in upregulation of *fng* transcripts in larval wings. Interestingly, *patched-Gal4*-driven overexpression of *fng* leads to an autonomous loss of wing margin marker expression as well as ectopic expression of wing margin marker on the posterior border of the Fng overexpressing region (*Panin et al., 1997*). All *Atro*[35] clones that cross the wing margin cause an autonomous loss of wing margin markers and large *Atro*[35] clones can induce some ectopic wing margin marker expression on the posterior edge of the clone, mimicking *patched-Gal4*-driven Fng overexpression. *Atro*[35] clones also disrupt N signaling reporter expression. *Atro*[35] clones cause the N signaling reporter to express diffusely instead of in a sharp line. This diffuse expression pattern may be an indication of a loss of precise N signaling at the wing margin. Interestingly, Fng is crucial for N signaling at the margin (*de Celis and Bray, 1997*; *Panin et al., 1997*). Thus, Atro represses *fng* expression, and loss of *Atro* and *patched-Gal4*-driven overexpression of Fng have similar N-related phenotype in the wing.

Loss of Atro also leads to Notch loss of function in the eye. *Atro*[35] clones have extra cells with the early R8 marker, Sens, indicating a defect in lateral inhibition. Although there are extra Sens-positive cells in *Atro* clones, not all these cells express the late R8 marker, Boss, thus the extra Sens-positive cells do not differentiate into R8. Inspection of the Sens-positive cell clusters reveals there is one cell with more Sens than its neighbors in each cluster. We reason that the difference in Sens levels renders one cell with the most R8-like and this cell can express Boss. *Atro*[35] clones also exhibit a loss of R7 and cone cell markers. Thus, Atro binds to the putative regulatory regions of genes that are connected to N signaling (such as *emc*, *Dl*, *mam*, *fng*, *neur*, *numb*, *Supplementary file 1*), genetically interacts with N and regulates N target expression.

While the strongest overlap in our ChIP-seq peaks was seen between Atro and Trl, we also observed significant overlap of our Atro ChIP-seq data with ChIP data sets of Yorkie (Yki), the key transcriptional co-activator of the Hippo pathway (Figure 5B). Interestingly, the atypical cadherin Fat regulates Yki activity via the Hippo pathway ([Bennett and Harvey, 2006; Cho et al., 2006; Silva et al., 2006; Willecke et al., 2006] and reviewed in Enderle and McNeill [2013] and planar polarity via Atro (Fanto et al., 2003; Saburi et al., 2012; Sharma and McNeill, 2013). Significantly, neurodegeneration by Atro has been shown to be mediated in part by Yki and the Hippo pathway (Napoletano et al., 2011). Thererfore, our finding that Yki and Atro are found at the same loci suggests a direct mechanism by which Atro may impact neurodegeneration, and suggests that Atro interactions with Yki may feed back into growth and patterning regulation by Fat cadherins.

To our knowledge, this is the first genome-wide analysis of Atrophin. Our genome-wide ChIP-seq and phenotypic analyses reveal many novel direct targets of Atro, and showed that Engrailed, Notch and Dpp signaling are directly regulated by Atro. Our analyses indicate that Atro preferentially binds to Trl binding sites. Interestingly, the fraction of paused genes is significantly more correlated with sites that bind both Atro and Trl, than just Trl alone, suggesting Atro may have a function in the regulation of pausing. Significantly, ChIP-re-ChIP experiments reveal that Atro and Trl bind to the same loci simultaneously, and phenotypic analyses indicate that Atro restricts expression of genes whose expression is promoted by Trl. Taken together, our data indicate that Atro is a critical component of developmental signaling and is an important general modulator of transcription activation by Trl.

## Materials and methods

### Double crosslink ChIP and subsequent ChIP-seq

Two step crosslinking ChIP was required in order for ChIP to enrich for positive controls when using the SG2524 anti-Atro antibody (see below). S2 cells ($10^7$ cells/mL, S2-DGRC, stock #6 from Drosophila Genomics Research Center) were washed three times in sterile PBS. Washed cells were first fixed with 2 mM ethylene glycol bis(succinimidyl succinate) (EGS) in PBS for 45 min at room temperature followed by three PBS washes. Washed cells were crosslinked in 1% formaldehyde in PBS for 15 min at room temperature and quenched in 125 mM glycine for 5 min on ice. Cells were then washed once in ChIP Wash Buffer A (10 mM Hepes pH7.6, 10 mM EDTA pH8.0, 0.5 mM EGTA pH8.0, 0.25% Triton X-100) and followed by a wash in ChIP Wash Buffer B (10 mM Hepes pH7.6, 100 mM NaCl, 1 mM EDTA pH8.0, 0.5 mM EGTA pH8.0, 0.01% Triton X-100) at 4°C, 5 min each. Washed cells were resuspended in Sonication buffer (50 mM Hepes pH7.6, 140 mM NaCl, 1 mM EDTA pH8.0, 1% Triton X-100, 0.1% sodium deoxycholate, 0.1% SDS, supplemented with proteinase inhibitors) at a concentration of $10^8$ cells/mL. Cells were then sonicated using a Qsonica Q700 sonicator in an ice-water bath (until fragments were roughly 150 bp in length). 10 μL of 10% SDS, 100 μL of 1% sodium deoxycholate, 100 μL of 10% Triton X-100, 28 μL of 5M NaCl were added to each mL of sonicated chromatin and incubated at 4°C for 10 min. Sonicated chromatin was then spun down at ≥20,000g for 5 min to remove cellular debris and the supernatant was used for ChIP.

Protein G Dynabeads (Invitrogen, Lithuania) were blocked in 1 mg/mL BSA in sonication buffer for at least 2 hr at 4°C. Blocked beads were conjugated with the antibodies for at least 4 hr at 4°C. 5 μL anti-Atro sera (SG2524, raised in rabbits against Atro amino acids 121–134, KGIDKKWTEDETKK), and 5 μL normal rabbit IgG (Cell Signaling Technology, Danvers, MA) were used for ChIP. Of the sonicated chromatin, 300 μL were incubated with conjugated beads overnight on a rotating wheel at 4°C. Beads were washed for 5 min each in Sonication buffer, Wash A (50 mM Hepes pH7.6, 500 mM NaCl, 1 mM EDTA pH8.0, 1% Triton X-100, 0.1% sodium deoxycholate, 0.1% SDS, supplemented with proteinase inhibitors), Wash B (20 mM Tris pH8.0, 1 mM EDTA pH8.0, 250 mM LiCl, 0.5% NP-40, 0.5% sodium deoxycholate), and TE buffer. Beads were then resuspended in Elution Buffer (50 mM Tris pH8.0, 50 mM NaCl, 2 mM EDTA, 0.75% SDS, 20 ug/mL RNase A) and incubated at 68°C overnight to remove crosslinks. Eluted chromatin was extracted by treating with Proteinase K followed by phenol chloroform DNA extraction (with 6 μg glycogen added during DNA precipitation [Thermo Scientific, Lithuania]). The extracted DNA was resuspended in 50 μL Tris pH8.0.

Three biological replicates were used for library construction (along with three corresponding IgG ChIP control). ChIP samples were treated with polynucleotide kinase and DNA polymerases for 30 min at room temperature (35 μL ChIP sample, 5 μL 10xNEB two buffer (New Englands Biolabs,

Ipswich, MA), 2 µL of 25 mM ATP, 2 µL of 10 mM dNTP, 10U T4 polynucleotide kinase, 4.5U T4 DNA polymerase, 1U Klenow Large Fragment DNA Polymerase, water to a volume of 50 uL). Afterwards, DNA was purified with PEG bead slurry (1M NaCl, 23% PEG, Sera-Mag Speedbeads (Fisher Scientific, England) with final PEG concentration of 13.87%) and eluted with 35 uL Qiagen EB buffer (Qiagen, Valencia, CA). Single dA overhang was added to eluted ChIP samples by incubating samples in 35 µL ChIP samples, 5 µL 10X NEB two buffer, 1 µL 10 mM dATP, 5U Klenow Fragment (3'−5' exo-) (New England Biolab), water to 50 µL for 30 min at 37°C. Samples were purified with PEG bead slurry and eluted with 35 µL Qiagen EB buffer. Short adapators for sequencing were ligated to samples by incubating samples at room temperature overnight in ligation buffer (35 µL DNA, 12 µL 5X quick ligation buffer, 2000U quick T4 DNA ligase, 2 µL 0.5 µM Illumina short sequencing adaptor, water to 60 µL). ChIP samples were purified two times with PEG bead slurry (started with 20% PEG instead of 23% PEG; final PEG of 8.89% and 10.91%, respectively) and eluted with 35 µL Qiagen EB buffer. Adaptor ligated libraries were PCR amplified (10 cycles), purified with PEG bead slurry (started with 20% PEG instead of 23%; final PEG concentration of 9.19%), and eluted with 35 µL Qiagen EB buffer. Libraries were then sequenced on Miseq (Illumina, San Diego, CA) using PE150V3 kit PE 85 bp. BWA program (v0.6.1; using default parameters) was used to align sequence reads to *Drosophila* genome release five (*Attrill et al., 2016*).

## Single crosslink ChIP

A second, independent ChIP was performed with a monoclonal antibody (4H6) raised against Atro amino acids 1369–1378 (SRQSPLHPVP) using *Drosophila* S2 cells catalog #006 from the *Drosophila* Genomics Resource Center. This ChIP was performed using two biological replicates (cells grown and ChIP'ed at different times). The cells were grown to a density of $0.2–1 \times 10^7$ cells/mL and fixed in 1% formaldehyde for 15 min at ambient temperature. The reaction was quenched by 0.16 M glycine pH 7.0 for 5 min and washed in PBS. Cells were sequentially washed with ChIP Wash buffer A and ChIP Wash Buffer B for 10 min at 4°C followed by resuspension in Sonication buffer to a final concentration of $5–10 \times 10^7$ cells/mL. Nuclei were sonicated for 15 min using a Diagenode Bioruptor, rotated for 10 min followed by centrifugation for 10 min at 13,000 rpm at 4°C.

A mix of Protein A and G Dynabeads (Invitrogen, Lithuania) blocked with BSA (Sigma Aldrich, St Louis, MO) were mixed with the antibody. Beads and antibodies were incubated for at least 2 hr followed by the addition of $0.5–1 \times 10^7$ cells. Chromatin and antibody bead complexes were formed during at least 2 hr followed by 5 min washes with Sonication buffer, Wash A, Wash B and TE buffer. Beads were resuspended in Elution buffer (same as above but supplemented with 20 µg/mL glycogen) in a new tube. Cross-linking was reversed at 68°C for at least 4 hr and proteins removed by Proteinase K digestion. DNA was purified by phenol-chloroform extraction, ethanol precipitated and finally resuspended in 200 µl 0.1×TE.

The DNA was sequenced at the Uppsala Genome Center using SOLiD (TM) ChIP-Seq Library preparation, size selection (around 150 bp + adapters 95 bp) and sequenced using SOLiD4 75 bp fragment run. The number of mapped reads were 11270731 (Input 1), 13320338 (Atro ChIP 1), 7315911 (Input 2) and 6972016 (Atro ChIP 2).

## ChIP-re-ChIP

S2 cells were double crosslinked, sonicated, and ChIP'ed as above, using 5 µL anti-Atro sera (SG2524), and 10 µL normal rabbit IgG (Cell Signaling Technology). After the first ChIP, beads were washed and eluted for re-ChIP as described in *Truax and Greer (2012)*. Eluted chromatin was incubated in BSA-blocked Dynabeads for 2 hr to remove any leftover antibodies. The supernatant containing eluted chromatin was re-ChIP'ed by incubating in beads conjugated with the appropriate amount of antibodies overnight (5 µL anti-Atro sera (SG2524), 10 µL anti-Trl sera (gift from K. White), 10 µL normal rabbit IgG (Cell Signaling Technology)). After the re-ChIP, the beads were washed and eluted as normal double crosslink ChIP samples above. qPCRs were done in technical triplicates with SYBR green PCR Master Mix (Applied Biosystems, Canada, 20 µL reaction volume; qPCR was performed three times for each ChIP samples). Percent input and errors were calculated using standard percent input calculations. qPCR primer pairs are listed in *Supplementary file 2*.

## Bioinformatics

For each biological replicate, peaks were called using MACS2 (version 2.1.0, FDR = 0.01, genome size $1.2 \times 10^8$). Each Atro ChIP-seq replicate was compared with its corresponding IgG ChIP-seq replicate control (or input control for the second independent Atro ChIP-seq replicates) in the MACS analysis. Peaks from the biological replicates were intersected and only peaks present across all replicates with summits within 100 bp were selected. Peaks were extended by 2 kb on both sides and then annotated by intersecting all *Drosophila* genes' coordinates with the peaks coordinates using BEDTools (*Quinlan and Hall, 2010*). MEME-ChIP was carried out using first-order model, any number of repetitions, motif count of 10, score ≥5 and an E-value threshold of ≤10. Overlap of Atro ChIP with Trl ChIP data sets (*Figure 5C*) was done using BEDTools Intersect function. Atro-binding sites were compared to data from modENCODE (downloaded from http://intermine.modencode.org/), (*Philip et al., 2015*) for CBP, and (*Oh et al., 2013*) for Yki, and gene expression divided into three equally sized bins (low, medium and high expressions) from (*Cherbas et al., 2011*) using custom Perl scripts.

## Principal component analyses

The enrichment values for each factor in the Atro binding regions were calculated by taking the mean of the top three consecutive enrichment values within each region (*Philip et al., 2015*). All factors were normalized so that 0 represents the genomic mean (background levels) and one represents the genomic maximum (mean of top 0.001 percentile) for each factor. Enriched Atro regions were used as observations and normalized enrichment values of each factor within the regions were used as variables as decribed in (*Philip et al., 2015*). Hierarchical clustering was done on all significant components of the analysis using Ward clustering to calculate tree distances. The three classes of Atro-bound regions were based on hierachical clustering.

## Clones and immunofluorescence

Mitotic clones were generated in flies with the following genotypes:

hs-flp; ; Atro³⁵ FRT80B / ubi-GFP FRT80B
hs-flp; ; Trl^R85 FRT2A / hs-GFP, Minute, FRT2A
hs-flp; NRE-GFP / +; Atro³⁵ FRT80B / arm-LacZ FRT80B

Mitotic clones were generated by heat shocking (at 37°C) larvae for 45 min at 48 hr and 72 hr after egg laying. Trl^R85 clones were additionally heat shocked for 45 min about 1.5 hr prior to dissection to induce GFP expression. Larval tissues were dissected and prepared as in standard protocol. Sample sizes are listed in *Supplementary file 3*. The following antibodies were used for immunofluorescence: mouse antibodies against En (DSHB 4D9, 1/400), Pros (DSHB MR1A, 1/500), Ct (DSHB 2B10, 1/500), Wg (DSHB 4D4, 1/500), and ß-Gal (Promega [Madison, WI], 1/1000); rabbit antibodies against En (Santa Cruz (Santa Cruz, CA) d-300, 1/500), Omb (gift from G. Pflugfelder, 1/1000), Dpp (gift from M. Gibson, 1/100), Ttk (gift from W. Ge, 1/100), and pMad (Cell Signaling Technology (Danvers, MA) #9510, 1/500); guinea pig antibodies against Sens (gift from H. Bellen, 1/1000), and Runt (gift from C. Desplan, 1/500).

## In situ hybridization

*enGal4 / +; UAS Atro RNAi* (Bloomington stock center, line 32961) / + larvae were used for in situ hybridization. Larval wing discs are prepared and stained as described in *Morris et al. (2009)*, substituting wing discs for testes. Primers for probes are listed in *Supplementary file 2*.

## S2 cells and knockdown

S2 cells were purchased from *Drosophila* Genomics Resource Center (S2-DGRC, stock #6). S2 cells were grown in S2 media + 10% fetal bovine serum in standard conditions, and cells were kept healthy for all experiments. Double stranded RNA was made using the Megascript T7 kit (Life Technologies, Canada). Primers used to make the dsRNA are listed in *Supplementary file 2*. S2 cells were dsRNA treatment using standard bathing S2 knockdown protocol. 10 ug/mL of dsRNA was used for each treatment.

## Immunoprecipitation

S2 cells ($10^7$ cells per immunoprecipitation) were incubated in 0.5 mL LPC buffer (5% sucrose, 35 mM Hepes pH7.4, 80 mM KCl, 5 mM $K_2HPO_4$, 5 mM $MgCl_2$, 5 mM $CaCl_2$, 0.01% α-lysophospatidyl-choline) for 3 min at room temperature. Cells were resuspended in 0.2 mL MNase buffer (5% Glyc-erol, 20 mM Tris pH7.4, 60 mM KCl, 15 mM NaCl, 5 mM $CaCl_2$, 3 mM $MgCl_2$, 0.5% NP-40, 1 mM DTT). 6 μL of micrococcal nuclease (40 U/μL, New England Biolabs) were added and incubated at 25°C for 5 min. 0.3 mL of MNase dilution buffer (3.6 mM Tris pH8.8, 12 mM EDTA, 225 mM NaCl, 60 mM KCl, 1.2% NP-40, supplemented with proteinase inhibitor) were added and incubated on shaker for 10 min at 4°C. MNase treated samples were centrifuged to remove cellular debris. Super-natant were used for immunoprecipitation using standard methods.

## Accession number

ChIP-Sequencing data reported in this study are archived at the Gene Expression Omnibus (https://www.ncbi.nlm.nih.gov/geo/) as accession numbers GSE87509 (ChIP using Atro antibody SG2524) and GSE87471 (second independent Atro ChIP with antibody 4H6).

## Acknowledgements

We thank Hugo Bellen, Sarah Bray, Gerrard Campbell, Claude Desplan, Maxim Frolov, Wanzhong Ge, Matthew Gibson, Judy Kassis, Henry Krause, John Lis, Gert Pflugfelder, Kevin White and Jeff Wrana for antibody reagents and fly stocks; Tim Hughes, Hamed Shateri Najafabadi and Tim West-wood for ChIP analyses help. This work was supported by grants from the Candian Institutes of Health Research Foundation (FDN 143319) and a Terry Fox Research Institute Team Grant to HM. HM is a Tier 1 Canada Research Chair in Coordinating Growth and Polarity. This work was also partly supported by grants from the Swedish Research Council to MMannervik, the UK Medical Research Council (NIRG-G1002186) to MF, and Knut and Alice Wallenberg to EpiCoN, co-PI: PS.

## Additional information

### Competing interests

HM: Reviewing editor, *eLife*. The other authors declare that no competing interests exist.

### Funding

| Funder | Grant reference number | Author |
| --- | --- | --- |
| Canadian Institutes of Health Research | FDN 143319 | Helen McNeill |
| Medical Research Council | NIRG-G1002186 | Manolis Fanto |
| Knut och Alice Wallenbergs Stiftelse | | Per Stenberg |
| Terry Fox Research Institute team grant | | Helen McNeill |

The funders had no role in study design, data collection and interpretation, or the decision to submit the work for publication.

### Author contributions

KY, Conceptualization, Formal analysis, Validation, Investigation, Visualization, Methodology, Writ-ing—original draft, Writing—review and editing; AB, Data curation, Formal analysis, Investigation; EK, Software, Formal analysis, Investigation, Methodology; P-HH, YT, IN, Formal analysis, Investiga-tion; DY, Resources, Methodology; AL, MMo, Resources, Data curation, Software, Methodology; MH, Resources, Data curation, Software, Supervision, Writing—review and editing; SA, Resources, Data curation, Supervision, Funding acquisition, Methodology; MF, Formal analysis, Supervision, Funding acquisition, Investigation, Writing—review and editing; PS, Software, Formal analysis, Supervision, Funding acquisition, Methodology, Writing—review and editing; MMa, Resources,

Formal analysis, Supervision, Investigation, Writing—review and editing; HM, Conceptualization, Formal analysis, Supervision, Funding acquisition, Investigation, Visualization, Writing—review and editing

Author ORCIDs

Kelvin Yeung, http://orcid.org/0000-0003-2284-8734
Damian Yap, http://orcid.org/0000-0002-5370-4592
Helen McNeill, http://orcid.org/0000-0003-1126-5154

## Additional files

### Supplementary files

• Source code 1. Script used to calculate overlap between data sets as Atro binding increases. Perl script used to count the frequency of overlap between various data sets in gff3-format to a reference gff3 file. Each data set is divided into a user defined number of bins by equally distributed cut-offs by value or by rank. Alternatively, the bins can be created so that all data points below each cut-off are merged into one bin. The frequency of overlap between the reference file data to the data points in each bin is reported.

• Source code 2. Script used to calculate overlap between data sets as Atro binding increases. Perl script used to count the frequency of overlap between various data sets in sgr-format to a reference gff3 file. Each data set is divided into a user defined number of bins by equally distributed cut-offs by value or by rank. Alternatively, the bins can be created so that all data points below each cut-off are merged into one bin. The frequency of overlap between the reference file data to the data points in each bin is reported.

• Source code 3. Perl script used to calculate the mean enrichment value of data points in a sgr-file falling within each region in a reference gff3-file. The user defines the number of data points within each region to include in the mean. The mean can be reported based on all values, the $n^{th}$ top values or the highest n consecutive values.

• Supplementary file 1. Table showing the list of potential target genes of Atro as identified by proximity to Atro ChIP peaks.

• Supplementary file 2. List of primers used for dsRNA, qPCR and in situ hybridization experiments.

• Supplementary file 3. Sample size information for clonal analyses.

### Major datasets

The following datasets were generated:

| Author(s) | Year | Dataset title | Dataset URL | Database, license, and accessibility information |
|---|---|---|---|---|
| McNeill H, Yeung K | 2017 | ChIP-seq of Atrophin in Drosophila S2 cells | https://www.ncbi.nlm.nih.gov/geo/query/acc.cgi?acc=GSE87509 | Publicly available at the NCBI Gene (accession no: GSE87509) |
| Boija A, Stenberg P, Mannervik M | 2017 | Atrophin (Atro) ChIP-seq data from Drosophila S2 cells | https://www.ncbi.nlm.nih.gov/geo/query/acc.cgi?acc=GSE87471 | Publicly available at the NCBI Gene (accession no: GSE87471) |

The following previously published datasets were used:

| | | | | Database, license, and accessibility |
|---|---|---|---|---|

| Author(s) | Year | Dataset title | Dataset URL | information |
|---|---|---|---|---|
| White K | 2010 | Chromatin Binding Site Mapping of Transcription Factors in D. melanogaster by ChIP-seq | http://intermine.moden-code.org/query/experi-ment.do?experiment=Chromatin+Binding+Site+Mapping+of+Tran-nscription+Factors+in+D.+melanogaster+by+ChIP-seq | Publicly available at Modencode (http://intermine.modencode.org) |
| White K | 2009 | Chromatin Binding Site Mapping | http://intermine.moden-code.org/query/experi-ment.do?experiment=Chromatin+Binding+Site+Mapping | Publicly available at Modencode (http://intermine.modencode.org) |
| Fuda NJ, Guertin MJ, Sharma S, Danko CG, Martins AL, Siepel A, Lis JT | 2013 | The Genomic Binding Profile of GAGA Element Associated Factor (GAF) in Drosophila S2 cells | https://www.ncbi.nlm.nih.gov/geo/query/acc.cgi?acc=GSE40646 | Publicly available at the NCBI Gene Expression Omnibus (accession no: GSE40646) |

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
