## [Decision Letter]

Thank you for submitting your article "Atrophin controls developmental signaling pathways via interactions with Trithorax-like" for consideration by *eLife*. Your article has been reviewed by two peer reviewers, and the evaluation has been overseen by K VijayRaghavan as the Senior Editor and Reviewing Editor. The reviewers have opted to remain anonymous.

The reviewers have discussed the reviews with one another and the Reviewing Editor has drafted this decision to help you prepare a revised submission.

Summary and Essential revisions:

This is an interesting paper that uses an analysis of the genome-wide binding sites of the co-repressor Atrophin in *Drosophila* to increase understanding of target genes under Atrophin control and to suggest a mechanism of Atrophin function as a cofactor regulating the activity of Trithorax-like (Trl). This is a valuable addition to our understanding of how signaling pathways are regulated.

Based on Atrophin binding peaks close to developmental pathway genes the authors explore the role of Atrophin in a number of pathways through analysis of Atrophin mutant clones. Evidence is presented implicating Atrophin as a regulator of engrailed, the Dpp pathway and Notch signalling. The binding data allow the inference of direct targets and this study is more comprehensive than prior analyses although Atrophin has been previously implicated in the regulation of engrailed and Dpp signalling.

Motif analysis on the Atrophin binding peaks indicates a strong enrichment for GA repeats, a motif associated with Trl. The majority of Atrophin binding peaks overlap Trl binding peaks and sequential ChIP experiments support the simultaneous binding of both Atrophin and Trl. Functional experiments on Atrophin targets, engrailed and thick veins, indicate that they are also regulated by Trl. All this forms an association between Atrophin and Trl and supports the authors' model that Atrophin can act to modulate gene activation by Trl.

The question is then how close is the association between Atrophin and Trl? The authors test this by co-IP and report no evidence for interaction in conventional extracts, however they do show co-IP following chromatin solubilisation using micrococcal nuclease. The authors conclude that Atrophin and Trl participate in a complex and suggest that Atrophin is a major Trl cofactor. Whilst there is good evidence for a co-localisation of Atrophin and Trl on chromatin the evidence for direct interaction seems weak. It remains plausible that Trl is associated with open chromatin regions where Atrophin may be recruited by a number of transcription factors and it is not clear that a direct interaction with Trl is required.

Although the association between Atrophin and Trl, both at the binding and functional levels, is interesting we do not think sufficient evidence is provided for the involvement of Atrophin in a clear mechanism as a direct cofactor of Trl. The discussions should appropriately be toned-down.

This paper is relatively straightforward and presents a consistent picture about the role of Atrophin in transcriptional regulation. The data are of high quality and convincing and they support the authors' conclusions. Overall this is a valuable and significant addition to the field.

*Reviewer #1 (Minor Comments):*

1) Figure 1. How are the 1275 S1-3-5 peaks derived from the 1377 common peaks mentioned in the text?

2) Subsection “Atro modulates developmental gene expression during larval development”, first paragraph: "two strong Atro peaks" upstream of the en TSS are described as being "in the en promoter". What are the limits of the en promoter and has it been defined? Similarly, subsection “Atro directly regulates Dpp signaling via *thickveins*”: "Atro bind's tkv's regulatory region"; what is this region and has it been defined? Also similarly, subsection “Atro is required for Notch signaling and directly regulates Notch pathway gene expression”, second and last paragraphs: "Atro binds to the regulatory region of fng" and "regulatory regions of multiple N signaling components"; what is the evidence these are regulatory regions?

3) The clonal analysis data is not always compelling; particularly the enhancement in Figure 2' is difficult to see in the image provided.

4) As the suggestion of a direct complex between Atro and Trl seems to me to be a major, or the major, point of interest in the paper, why is the critical co-IP experiment in a supplementary figure?

5) Figure 5. Although most Atro peaks overlap Trl peaks, a sizeable minority do not. Is this just a genomic data thresholding issue or is there good evidence for Atro binding in the absence of Trl? The peak the authors use in Figure 6 seems to be a clear example but it would be good if the authors could comment in general on the Atro sites that lack Trl and how they fit in their model.

*Reviewer #2 (Minor Comments):*

1) The choice about which data to include in the supplemental figures sometimes seems odd. For example, the co-IP data seem central to their conclusions and so I don't understand why they are relegated to a supplemental figure.

2) Perhaps it's there somewhere, but I can't readily find the source of the genomic data that was used to generate the graph shown in Figure 5.

3) Does bioinformatic analysis give any clue as to the difference between the Trl peaks that bind Atro and those that do not?

4) Given that the co-IP analysis was performed on MNase treated crude lysates, the interaction between Atro and Trl could be very indirect. Do the authors think it depends on the presence of chromatin fragments in the lysates?

5) Some of the ChIP experiments used double cross-linking. Is this because they found that the double cross-linking enhanced their signal?

---

## [Author Response]

*[…] Reviewer #1 (Minor Comments):*

*1) Figure 1. How are the 1275 S1-3-5 peaks derived from the 1377 common peaks mentioned in the text?*

The 1275 peaks in Figure 1 are only the peaks in the major chromosome arms from the 1377 common peaks. The peaks in the heterochromatin regions (e.g. chr3Rhet) are excluded, hence there are less peaks in Figure 1 than the common 1377 peaks. This has been clarified in the main text and in the figure legends.

*2) Subsection “Atro modulates developmental gene expression during larval development”, first paragraph: "two strong Atro peaks" upstream of the en TSS are described as being "in the en promoter". What are the limits of the en promoter and has it been defined? Similarly, subsection “Atro directly regulates Dpp signaling via thickveins”: "Atro bind's tkv's regulatory region"; what is this region and has it been defined? Also similarly, subsection “Atro is required for Notch signaling and directly regulates Notch pathway gene expression”, second and last paragraphs: "Atro binds to the regulatory region of fng" and "regulatory regions of multiple N signaling components"; what is the evidence these are regulatory regions?*

Corresponding sources for promoter and regulatory gene regions have now been cited in the main text. For genes whose regulatory regions were not formally defined, these were changed to near putative regulatory regions of potential target genes instead of regulatory regions.

*3) The clonal analysis data is not always compelling; particularly the enhancement in Figure 2' is difficult to see in the image provided.*

In the wing, endogenous En is expressed very strongly in the wing pouch. Hence, it is difficult to see a clear upregulation caused by *Atro^35^* clones. To counter this problem, we generated new *Atro^35^* wing clones and visualized En using weaker fluorescent secondary antibody to show a clearer upregulation of En. Accordingly, this new figure replaced the old Figure 2.

*4) As the suggestion of a direct complex between Atro and Trl seems to me to be a major, or the major, point of interest in the paper, why is the critical co-IP experiment in a supplementary figure?*

We agree with this point, which was brought up by both reviewers 1 and 2. In response to their comment, we have moved the coIP experiment to the main figure (Figure 5).

*5) Figure 5. Although most Atro peaks overlap Trl peaks, a sizeable minority do not. Is this just a genomic data thresholding issue or is there good evidence for Atro binding in the absence of Trl? The peak the authors use in Figure 6 seems to be a clear example but it would be good if the authors could comment in general on the Atro sites that lack Trl and how they fit in their model.*

We believe that Atro can interact with other transcription factors other than Trl to associate to DNA. For example, Atro was shown to physically interact with Tailless and Cubitus Interruptus (Wang et al., 2006; Zhang et al., 2013). Thus this may explain why some Atro peaks do not intersect with Trl.

In addition to the above, we also believe that some Atro peaks that do not intersect with Trl in our analysis are weaker than the peaks that do. In order to test this we used the data generated by MACS2 peak calling:

MACS2 provides a –log(pvalue) value for each corresponding peak. This value ranks the significance of each of the peaks, where MACS2 assigns a higher –log(pvalue) for highly significant peaks. Using this data, we plotted box plots for all Atro peaks that intersect Trl and all Atro peaks that do not intersect with Trl, and then compared the two. The plots show that Atro peaks that do not intersect with Trl have slightly less –log(p-value) than that of Atro peaks that intersect Trl (Wilcoxon test p-value < 0.001). MACS2 also generated a fold-enrichment value for each of the called peaks and we plotted box plots using these values as well. We saw that Atro peaks that do not intersect with Trl have less fold enrichment than Atro peaks that do (Wilcoxon test p-value < 0.001). These data suggest that the Atro peaks that do not overlap with Trl are less significant than those that intersect.

Therefore, we believe that the Atro peaks that do not intersect Trl can be explained by: 1) Atro interacting with factors other than Trl and 2) Atro peaks that do not intersect with Trl tend to be less significant than the peaks that do intersect. The above points have been added to the main text and the plots have been added as Figure 5—figure supplement 2.

*Reviewer #2 (Minor Comments):*

*1) The choice about which data to include in the supplemental figures sometimes seems odd. For example, the co-IP data seem central to their conclusions and so I don't understand why they are relegated to a supplemental figure.*

We agree with the reviewers regarding this and in response, we have moved the co-IP figure to the main figure (Figure 5).

*2) Perhaps it's there somewhere, but I can't readily find the source of the genomic data that was used to generate the graph shown in Figure 5.*

The sources of the genomic data are from modencode.org, and Philip et al., 2015 and Oh et al., 2012 for CPB and Yki, respectively. Only Atro peaks from major chromosome arms (e.g. heterochromatin excluded) were used for this figure. This information has been added to the main text as well as in the figure legends.

*3) Does bioinformatic analysis give any clue as to the difference between the Trl peaks that bind Atro and those that do not?*

To address this, we repeated our bioinformatics analyses on Atro for Trl peaks. We first looked at the distribution of Trl peaks that do not intersect with Atro and Atro peaks that intersect with Trl to verified genomic regions (e.g. promoters, introns). However, we could not see any obvious differences between the two sets of data.

We also performed principal component analysis (PCA) of Atro peaks and the PCA indicated that Atro peaks can be grouped into 3 different classes. The PCA is now included as a figure supplement (Figure 1—figure supplement 1). We note that more experiments are needed to reveal the biological significance of these 3 groups.

*4) Given that the co-IP analysis was performed on MNase treated crude lysates, the interaction between Atro and Trl could be very indirect. Do the authors think it depends on the presence of chromatin fragments in the lysates?*

We agree with the reviewer that the interaction could be indirect and we have toned down our discussion accordingly. Currently, we do not know if the Atro-Trl interaction requires the presence of chromatin but the requirement of MNase to see a coIP of Atro and Trl suggests that Atro and Trl are interacting near chromatin.

*5) Some of the ChIP experiments used double cross-linking. Is this because they found that the double cross-linking enhanced their signal?*

Double cross-linking was crucial for the ChIP with the SG2524 anti-Atro antibody. Without double cross-linking, the Atro ChIP could not enrich for positive controls during the ChIP optimization. This point has been clarified in the Materials and methods section.